# The influence of metformin transporter gene *SLC22A1* and *SLC47A1* variants on steady-state pharmacokinetics and glycemic response

Vitarani Dwi Ananda Ningrum[1]*, Ahmad Hamim Sadewa[2], Zullies Ikawati[3], Rika Yuliwulandari[4], M. Robikhul Ikhsan[5], Rohmatul Fajriyah[6]

1 Department of Pharmacy, Universitas Islam Indonesia, Sleman, Daerah Istimewa Yogyakarta, Indonesia, 2 Faculty of Medicine, Universitas Gadjah Mada, Sleman, Daerah Istimewa Yogyakarta, Indonesia, 3 Faculty of Pharmacy, Universitas Gadjah Mada, Sleman, Daerah Istimewa Yogyakarta, Indonesia, 4 Faculty of Medicine, Universitas YARSI, Jakarta Pusat, Indonesia, 5 Department of Internal Medicine, Dr. Sardjito General Hospital, Sleman, Daerah Istimewa Yogyakarta, Indonesia, 6 Department of Statistics, Universitas Islam Indonesia, Sleman, Daerah Istimewa Yogyakarta, Indonesia

* vitarani.ningrum@uii.ac.id

**Data Availability Statement:** The authors cannot provide the subject data set since it contains confidential information on the patient's genetic

## Abstract

Interindividual variation is important in the response to metformin as the first-line therapy for type-2 diabetes mellitus (T2DM). Considering that OCT1 and MATE1 transporters determine the metformin pharmacokinetics, this study aimed to investigate the influence of *SLC22A1* and *SLC47A1* variants on the steady-state pharmacokinetics of metformin and the glycemic response. This research used the prospective-cohort study design for 81 patients with T2DM who received 500 mg metformin twice a day from six primary healthcare centers. *SLC22A1* rs628031 A>G (Met408Val) and Met420del genetic variants in OCT1 as well as *SLC47A1* rs2289669 G>A genetic variant in MATE1 were examined through the PCR-RFLP method. The bioanalysis of plasma metformin was performed in the validated reversed-phase HPLC-UV detector. The metformin steady-state concentration was measured for the trough concentration ($Css^{min}$) and peak concentration ($Css^{max}$). The pharmacodynamic parameters of metformin use were the fasting blood glucose (FBG) and glycated albumin (GA). Only *SLC22A1* Met420del alongside estimated-glomerular filtration rate (eGFR) affected both $Css^{max}$ and $Css^{min}$ with an extremely weak correlation. Meanwhile, *SLC47A1* rs2289669 and FBG were correlated. This study also found that there was no correlation between the three SNPs studied and GA, so only eGFR and $Css^{max}$ influenced GA. The average $Css^{max}$ in patients with the G allele of *SLC22A1* Met408Val, reaching 1.35-fold higher than those with the A allele, requires further studies with regard to metformin safe dose in order to avoid exceeding the recommended therapeutic range.

## Introduction

The incidence of diabetes mellitus (DM) in Indonesia is getting higher every year, reaching 2.1% increase since 2013 based on the 2018 National Basic Health Research report. Of the total population, 13.1% has a high level of fasting blood glucose [1]. Consequently, to prevent and

profile, and because of national policy. Should such data be needed, a request can be addressed to the Ethics Committee of the Faculty of Medicine of Universitas Gadjah Mada, Radiopoetro Building 2nd floor, Farmako, Sekip Utara St, Yogyakarta 55128, +62811-2666-869, e-mail: mhrec_fmugm@ugm.ac.id.

**Funding:** This study was partially funded by Universitas Islam Indonesia through the 2021 Research Grant No. 3355/Rek/10/DSDM/IX/2021 which covered only the funding of research materials.

**Competing interests:** The authors have declared that no competing interests exist.

decrease DM-induced mortality and morbidity, a good blood glucose management is needed [2].

During the period of 2013–2021, metformin had been listed in the Indonesian National Formulary along with other oral antidiabetic drugs, including glipizide, glimepiride, and glibenclamide, as a drug provided by primary healthcare centers [3]. Compared to other oral antidiabetic drugs, metformin has a better ability to decrease the level of HbA1c by 1.0–2.0% and has less hypoglycemia effects. However, it is known that the glycemic response to metformin is varied because 35–40% patients have not reached the target for fasting blood glucose [4]. Our previous research revealed high variability in metformin plasma steady-state concentration (PSSC), reaching >100x at the trough and 15x at the peak [5]. Genomic variation likely leads to patients' variability in the drug pharmacokinetic and pharmacodynamic variability, including those of metformin [6]. Metformin has renal excretion as the major elimination pathway with >0.6 genetic component (rGC), indicating that genetic factor greatly affects the variability in metformin renal clearance [7]. Genetic variation has an influence on the protein function in metformin bioavailability or therapeutic effects.

With the hydrophilic property as a cationic species (>99.9%) at a physiological pH, the pharmacokinetics of metformin is effective depending the function of the transporters [8]. The main transporters that have a key role in the pharmacokinetics of metformin to date are Organic Cation Transporter1 (OCT1) and Multidrug and Toxin Extrusion1 (MATE1). Mainly expressed in the liver, OCT1 is a protein transporter that carries metformin to hepatocytes, the target of metformin action. The genetic variation in *SLC22A1* as the OCT1 coding gene can change the protein function, leading to a reduced amount of metformin in the receptors and therefore a declined therapeutic response. A number of studies showed that *SLC22A1* genetic variation resulted in varied steady-state concentration of metformin and various glycemic response [9–12]. Furthermore, latest studies found that such genetic variation was associated with metformin intolerance in the gastrointestinal tract [13,14].

In addition, the *SLC47A1* is a MATE1 protein-coding gene mostly located in the apical membrane of renal tubular cells and canalicular membrane of hepatocytes. MATE1 transports metformin from hepatocytes to the bile and excretes metformin through the kidneys. Some research proved that the polymorphisms in *SLC47A1* affect the pharmacokinetic variability as well as the glycemic response [15,16]. To date, however, the majority of metformin pharmacogenetic studies focus on the effects of OCT1 and MATE1 polymorphisms on glycemic control at various doses. Only one study has linked this to the minimum steady-state concentrations but not to the maximum [10], which is likely associated with a predisposition to lactic acidosis. Meanwhile, a large number of studies of the peak concentrations only focus on single administration of metformin to healthy volunteers for bioavailability-bioequivalence studies but not on repeated administration as an actual condition of metformin use among T2DM patients. Both transporters are known to play an important role in metformin bioavailability. In addition, metformin pharmacogenetic studies conducted prospectively in a similar dose with a control on the adherence factors remain extremely limited. Therefore, this prospective study aimed to analyze how the genetic variation in two metformin transporter encoding genes correlates with not only the glycemic response but also with the minimum and maximum steady-state concentrations.

## Materials and methods

### Recruitment of the subjects

T2DM patients administered metformin 500 mg twice daily for at least 2 weeks from six primary healthcare centers in Yogyakarta Special Province were involved. An explanation of the

research, such as the objectives, the procedures for the participants to follow as well as the risks and benefits of the research were conveyed both orally and in writing directly to the eligible subjects. The subjects were allowed time to decide whether they would participate in the study. When they have verbally expressed their consent, they signed 2 (two) informed consent forms containing the consent to participate in the study (sheet 1) and to permit the research team to store and use their remaining specimens or DNA (sheet 2). The subjects recruited were in the 30–60 age range and literate, thus requiring no parent or guardian involvement in the subject recruitment procedure to indicate their consent to participate in this study. The ethical clearance was approved by the Ethics Committee of the faculty of Medicine of Universitas Gadjah Mada with the approval letter Number KE/FK/648/EC and conducted in accordance with the Declaration of Helsinki.

## Analysis of the genotypes

The genotype analysis was done through Polymerase Chain Reaction (PCR) and Restriction Fragment Length Polymorphism (RFLP).

*SLC22A1* **(OCT1) rs628031 (Met408Val).**   The PCR primer design used 5'-`TTT CTT CAG TCT CTG ACT CAT GCC`-3' and 5'-`AAA AAA CTT TGT AGA CAA AGG TAG CAC C`-3'. The analysis of the 397-bp amplification products was done in 1% agarose gel followed by the restriction digestion in *MscI* with 16–18 hours of incubation at 37˚C. The digestion yielded 397-bp fragments for the homozygous variants (val/val) as well as 210-bp and 187-bp fragments for the wild-type (Met/Met). The size of the digestion products (397 bp, 210 bp, and 187 bp) showed a category of heterozygotes (Met/Val). The genotype analysis were confirmed through the sequencing in a previous study [17].

*SLC22A1* **Met420del in OCT1.**   The PCR primer design used 5'-`AGGTTCACGGACTCT GTGCT`-3' as the forward primer and 5'-`AAGCTGGAGTGTGCGATCT`-3' as the reverse primer. The analysis of the 600-bp amplification products was done in 1% agarose gel at 100 Volt for 30 minutes, and the restriction digestion used *BspHI* with ±12 hours of incubation at 37˚C. The T-CATGA sequence was cut by the *BspHI* enzyme at 197th DNA template base. The *BspHI* identified and digested the AA genotype, but this enzyme did not identify the PCR products with a T-CATTT sequence, making such products remain undigested. The digestion produced 600-bp fragments of AA (wild type) genotype, 403-bp and 197-bp fragments of aa (mutant) genotype, and 600-bp, 403-bp, 197-bp of heterozygotes (Aa).

*SLC47A1* **(MATE1) rs2289669 (G>A).**   The PCR primer design used the forward primer of 5'-`TCA GTT TCC ACA GTA GCG TCG`-3' and the reverse primer of 5'-`GAC ACT GGA AGC CAC ACT GAA`-3'. The *TaqI* restriction endonuclease digested the amplification products (211 bp), which were then analyzed in 2% agarose gel. The restriction digestion used the *TaqI* with 16–18 hours of incubation at 65˚C. The 211-bp amplicons were digested into 21-bp and 190-bp fragments of AA genotype, 211-bp fragments of GG genotype/wild-type as well as 21-bp, 190-bp, and 211-bp fragments of heterozygous genotype [18].

## Pharmacokinetics of metformin steady-state concentrations

The patients reported the time of the last metformin administration which was done for uniform doses and intervals. In 12 hours after the last dose administration, they visited the primary healthcare center for blood sampling to measure the same-day trough and peak concentrations. The sampling for the trough PSSC was immediately done before the next-dose administration (pre-dose), while the peak PSSC sample was taken in 3.5–4.0 hours after the metformin administration (post-dose). The samples were then delivered to the Laboratory of Drugs, Food, and Cosmetics of the Pharmacy Department of Universitas Islam Indonesia to

be centrifuged for 10 minutes at 3500g, and the plasma aliquot was stored in a 2-ml polypropylene tube at -20°C in a maximum of one hour after the sampling. The metformin plasma concentrations were determined through a validated reversed-phase high performance liquid chromatography (HPLC) assay with Sunfire® C-18 column, 4.6 x 150mm x 5μm from Waters, and SM7 injector with an ultraviolet (UV) detector at 234 nm wavelength [19]. The metformin PSSC could estimate the elimination rate, and the metformin half-life was also calculated using the following formula [20].

$$K \ (/hour) = \frac{ln \left( \frac{Css^{max}}{Css^{min}} \right)}{8}$$

$$t_{1/2} \ (hour) = \frac{0,693}{K}$$

## Measurement of the glycemic response

The FBG and GA of T2DM patients given metformin monotherapy were measured before and after the continuous administration of metformin 500 mg twice daily for six weeks. The UV/VIS spectrophotometry of Hitachi 902® was used to measure FBG with the GOD-PAP method, and the ELISA reader of ADVIA® was employed in the measurement of GA with the KAOD (*Ketoamine oxidase*) method.

## Statistical analysis

The metformin PSSC obtained was displayed in mean ± SD values. A comparison of patients' metformin PSSC among the groups of allele types and genetic variants was made using the independent t-test and one-way ANOVA for normally distributed data as well as the Mann-Whitney and Kruskal-Wallis test for non-normal data distribution. To analyze the patient-related factors affecting the pharmacokinetics of metformin steady-state concentrations and glycemic control, the linier regression was employed with a statistically significant p value of ≤0.05.

## Results and discussion

There have been no prospective studies of the influence of genetic polymorphisms on the pharmacokinetics of steady-state concentrations and glycemic response that involve T2DM patients who adhere to metformin therapy with a similar dose for a minimum of eight weeks. Given that metformin is a long-term antidiabetic drug, the pharmacokinetic variability of repeated administration can give a more accurate description of the concentration variability, while in a single-dose administration it is left unknown.

The discussion on the effects of genetic polymorphisms on the variability of the pharmacokinetics of steady-state concentrations and glycemic control resulted from metformin use should begin with an understanding of the function, physiological role of OCT1 and MATE1 protein transporters, as well as the level of gene expression in various human tissues. The following table describes the predicted pharmacokinetic variability of metformin steady-state concentrations and its glycemic response with regard to SNPs in *SLC22A1* and *SLC47A1* genes.

Meanwhile, the research findings related to the steady-state pharmacokinetic variability in each genetic variant and allele of both target genes are presented in Table 1.

In general, Table 1 shows that the T2DM patients in the Javanese-Indonesian population have a significant difference in the Css^max between the Aa and aa variants. As previously described in Table 2, OCT1 is highly expressed in the basolateral membrane of hepatocytes,

**Table 1. Variability of metformin steady-state concentrations according to the genetic variants and alleles.**

| Group of Patients | Frequency (%) | Css<sup>min</sup> (µg/mL) (P Value) | Css<sup>max</sup> (µg/mL) (P Value) |
|---|---|---|---|
| *SLC22A1* Met408Val | | | |
| AA | 5 (6.17) | 0,358±0.292 | 0.818±0.445 |
| AG | 53 (64.43) | 0.365±0.244 | 1.323±0.854 |
| GG | 23 (28.40) | 0.347±0.335 | 1.006±0.654 |
| | | 0.964 | 0.144 |
| *SLC22A1* Met408Val | | | |
| A Allele (AA genotype) | 5 (6.17) | 0.596±0.486 | 1.363±0.743 |
| G Allele (AG and GG genotype) | 76 (93.83) | 0.600±0.453 | 1.845±0.944 |
| | | (0.986) | 0.265 |
| *SLC22A1* Met420del | | | |
| AA | 0 (0.00) | - | 2.831±0.518 |
| Aa | 3 (3.70) | 0.549±0.210 | 1.778±0.928 |
| Aa | 78 (96.30) | 0.352±0.272 | 0.015 |
| | | 0.222 | |
| *SLC47A1* rs2289669 | | | |
| GG | 14 (17.28) | 0.440±0.259 | 1.298±0.573 |
| GA | 35 (43.21) | 0.316±0.314 | 1.166±1.030 |
| AA | 32 (39.51) | 0.372±0.221 | 1.202±580 |
| | | 0.337 | 0.303 |
| *SLC47A1* rs2289669 | | | |
| G Allele (GG genotype) | 14 (17.28) | 0.720±0.435 | 1.846±0.727 |
| A Allele (GA and AA genotype) | 67 (82.72) | 0.574±0.455 | 1.810±0.978 |
| | | (0.277) | (0.618) |

Css<sup>max</sup>, maximum steady-state concentration; Css<sup>min</sup>, minimum steady-state concentration.

making the polymorphisms able to reduce the protein function of OCT1 in transporting metformin into hepatocytes as the action target. As a result, metformin is retained in the systemic circulation at a higher concentration than in wild-type patients. Since such variant was not found in this study, no further comparative analysis could be performed. Although the

**Table 2. Prediction of the steady-state pharmacokinetic variability and glycemic response affected by the genetic polymorphisms in *SLC22A1* and *SLC47A1*.**

| Location of *SNPs* | Affected stage of metformin pharmacokinetics | Prediction of the effects of SNPs on the glycemic control parameters (FBG and GA) based on metformin Css as opposed to that of the wild-type variant | | | | | |
|---|---|---|---|---|---|---|---|
| | | Css<sup>max</sup> | Css<sup>min</sup> | Final FBG value[a] | Changed FBG value[b] | Final GA value[a] | Changed GA value[b] |
| *SLC22A1* encoding OCT1 in the basolateral membrane of intestinal cells | Absorption | lower | minimum effect | higher | higher | higher | higher |
| *SLC22A1* encoding OCT1 in the basolateral membrane of hepatocytes* | Influx to the action target in hepatocytes | higher | minimum effect | higher | higher | higher | higher |
| *SLC47A1* encoding MATE1 in the hepatic canalicular membrane* | Efflux to the bile | lower | minimum effect | lower | lower | lower | lower |
| *SLC22A1* encoding OCT1 in the apical membrane of renal tubular cells | Reabsorption in the renal tubules | lower | Lower | higher | higher | higher | higher |
| *SLC47A1* encoding MATE1 in the brush-border membrane of renal tubular cells* | Efflux from the renal cells to be eliminated via urine | higher | higher | lower | lower | lower | lower |

Note

*highly expressed [31]; [a]after the administration of metformin 500 mg every 12 hours for 6 weeks.

[b]obtained from the final value of glycemic control (FBG, GA) minus the baseline value.

OCT1, organic cation transporter 1; MATE1, Multidrug and Toxin Extrusion 1; Css<sup>max</sup>, maximum steady-state concentration; Css<sup>min</sup>, minimum steady-state concentration; FBG, fasting blood glucose; GA, glycated albumin.

difference was insignificant, a finding similar to the prediction was indicated by the difference in the mean $Css^{max}$ between the A allele and the G allele in *SLC22A1* Met408Val, which was 1.363±0.743 g/mL and 1.845±0.944 g/mL, respectively. The presence of the rs628031 Met408-Val polymorphisms in *SLC22A1* is known to decrease the concentration of OCT1 mRNA in the human liver [21], resulting in reduced OCT1 function to transport metformin to hepatocytes. Consequently, polymorphisms in the *SLC22A1* gene decrease the function of OCT1 in transporting metformin to hepatocytes, resulting in the highest $Css^{max}$ being found in the variant-type group. In relation to the risk of lactic acidosis, the G allele has the higher potential than the A allele. Given the accumulation of metformin concentration becomes a predisposition to metformin associated lactic acidosis (MALA), the maximum recommended dose of metformin, particularly on the G allele, should be considered. A number of studies have found that metformin accumulation leads to lactatemia either with or without decreased renal function [22,23]. In fact, there is a 6-fold increased risk of lactic acidosis in the initial use of metformin alongside a decreased renal function, and a 12–13 times higher risk is found in patients with cumulative exposure to high-dose metformin in the past year or initial exposure to high-dose metformin [24]. The administration of subtherapeutic dose is not a solution since the glycemic target is by no means achieved [25]. Metformin dose is not correlated with plasma lactate or serum creatinine as shown in a study involving the incidence of MALA for over 30 years of observation [26]. There is no examination of metformin concentration or control of adherence factors, making the accumulation of metformin in plasma remain a predisposing factor for MALA. However, other factors such as BMI [27] and comorbidities, including renal impairment, also clearly become the co-factors of metformin accumulation to induce MALA [28–30]. Not only the dose but also the long-term metformin use become a risk factor for metformin accumulation due to the distribution of metformin into the erythrocyte compartment as previously found in our study [5].

Using an approach of elimination half-life calculation based on the allele type in *SLC22A1* Met408Val, this study found that the mean $Css^{max}$ was 1.35-fold higher in the G allele group (AG+GG) when compared to the wild-type group. A longer $t_{1/2}$ (1.25 times) was also found in the GG homozygous mutant group (Val/Val) when compared to the AG heterozygous group (Met/Val). Therefore, it is recommended that the maximum dose of metformin for patients with the G allele (AG+GG) is lower than that for the wild-type group, and a longer interval of administration is recommended for the GG homozygous mutant group (Val/Val) in order to minimize the incidence of lactic acidosis.

Meanwhile, the *SLC47A1* that encodes MATE1 is highly expressed in the canalicular membrane of hepatocytes in the bile and in the brush-border membrane of renal tubular cells. Each of which plays a role in the efflux to the bile and efflux from the kidney cells to be eliminated through urine, with predicted lower and higher $Css^{max}$ than those of the wild-type variant, respectively with the similar prediction for the glycemic response (Table 2). The mean metformin concentration in both the peak and trough PSSC is lower in the group of patients with the A allele of *SLC47A1* rs2289669 when compared to the wild-type group although there is no significant difference. The likely decreased function of MATE1 in the canalicular membrane of hepatocytes in the bile which is more significant that in the brush-border membrane of renal tubular cells requires further studies.

It is found in our previous study that differences in the regimen of oral antidiabetic drugs and the duration of metformin use have led to significantly different mean of $Css^{max}$ and $Css^{min}$, respectively. In addition, the linear regression analysis has shown that only the $Css^{max}$, alongside the glycemic control factors, affect FBG and GA while the $Css^{min}$ has an influence on FBG. Therefore, this study proceeds with a linear regression test to further analyze the patient factors, including the genetic variants in the two target genes that influence the steady-

**Table 3. Patient-related factors correlated with glycemic response after the administration of metformin 1000mg/day for 6 weeks.**

| Dependent variable | Predictor | Coefficient | Coefficient of correlation | P value | ANOVA test result | Adjusted R Square in the Model Summary |
|---|---|---|---|---|---|---|
| $Css^{min}$ (µg/mL) | eGFR | -0.006 | -0.246 | 0.026 | 0.015 | 0.093 |
| | Variant genotype of *SLC22A1* Met420del | -0.551 | -0.231 | 0.043 | | |
| | BMI | -0.020 | -0.200 | 0.080 | | |
| $Css^{max}$ (µg/mL) | eGFR | -0.013 | -0.258 | 0.018 | 0.009 | 0.103 |
| | Variant genotype of *SLC22A1* Met420del | -1.430 | -0.288 | 0.011 | | |
| | BMI | -0.029 | -0.135 | 0.228 | | |
| Metformin elimination half-life | Duration of previous metformin therapy | 3.696 | 0.254 | 0.022 | 0.029 | 0.064 |
| | Allele type of *SLC22A1* Met408Val | -4.542 | -0.181 | 0.101 | | |
| Final FBG | Baseline GA | 3.093 | 0.463 | 0.004 | 0.001 | 0.333 |
| | Variant genotype of *SLC47A1* rs2289669 | 20.460 | 0.404 | 0.011 | | |
| FBG change | Baseline FBG | 3.135 | 0.347 | 0.078 | 0.000 | 0.621 |
| | Baseline GA | -1.006 | -0.968 | 0.000 | | |
| | Variant genotype of *SLC47A1* rs2289669 | 20.425 | 0.299 | 0.014 | | |
| Final GA | Baseline GA | 1.142 | 1.274 | 0.000 | 0.000 | 0.727 |
| | Baseline FBG | -0.077 | -0.749 | 0.001 | | |
| | eGFR | 0.086 | 0.305 | 0.020 | | |
| | Variant genotype of *SLC47A1* rs2289669 | -0.889 | -0.131 | 0.222 | | |
| | $Css^{min}$ | -3.622 | -0.176 | 0.244 | | |
| | $Css^{max}$ | 2.582 | 0.443 | 0.011 | | |
| GA change | Baseline FBG | -0.793 | -0.838 | 0.000 | 0.000 | 0.460 |
| | eGFR | -0.060 | 0.353 | 0.039 | | |
| | Variant genotype of *SLC22A1* Met408Val | 0.068 | -0.192 | 0.182 | | |
| | $Css^{max}$ | 1.689 | 0.365 | 0.026 | | |

$Css^{max}$, maximum steady-state concentration; $Css^{min}$, minimum steady-state concentration; FBG, fasting blood glucose; GA, glycated albumin; eGFR, estimated-glomerular filtration rate; BMI, body mass index.

level pharmacokinetics, glycemic response, and time estimates for metformin elimination to provide an approximation of the effective metformin dose as presented in Table 3.

With regard to the use of glycemic control parameters, such as FBG and GA in this study but not HbA1c which is commonly used in the majority of metformin pharmacogenetic studies, a strong correlation between these three parameters has been demonstrated in some research [32–36]. The use of GA in Indonesia as both a diagnostic function and parameter for monitoring the success of diabetes therapy remains limited and has not become the gold standard of either the Indonesian Society of Endocrinology or the American Diabetic Association. However, GA is preferred for describing a glycemic control as opposed to HbA1c, especially in patients with impaired renal function or decreased life span of erythrocyte such as hemolytic anemia [37]. In addition, even with a shorter life span of albumin compared to that of HbA1c (±15 days), GA can describe the glycemic control in patients with diabetes mellitus for a

minimum of 2–3 weeks [38], making it more appropriate for this study which involves adherent patients taking metformin for eight weeks (including metformin use duration as the inclusion criteria).

Together with eGFR and BMI, the *SLC22A1* Met408del polymorphisms affect the pharmacokinetics of steady-state concentration of metformin with only a low adjusted R Square. This indicates that the two steady-state concentrations of metformin are mostly explained by other variables which are not involved in this study. Similarly, an extremely weak correlation is also shown by the *SLC22A1* Met408del variant type alongside the duration of previous metformin use and the elimination half-life of metformin. Meanwhile, in the glycemic response based on FBG, only *SLC47A1* rs2289669 affects both the decrease in and the final FBG values, particularly the decreased FBG with 0.621 adjusted R Square.

Encoded by the *SLC47A1* gene, the metformin transporter MATE1 is mostly expressed in the apical membrane of renal tubular cells and canalicular membrane of hepatocytes. It has therefore a major role in the final phase of cationic organic compound excretion, including metformin [39]. On the other hand, a number of studies have investigated *SLC47A1* rs2289669 polymorphisms and their effects on the pharmacokinetics, response, and other biochemical parameters for metformin. The rs2289669 polymorphisms interact with *SLC22A1* rs594709, thus decreasing FBG and postprandial insulin as well as increasing HOMA-IR in the AA genotype group that has *SLC22A1* and *SLC47A1* as opposed to the group with a G allele [40]. Therefore, this study confirms the correlation between rs2289669 *SLC47A1* polymorphisms and FBG values.

Meanwhile, when the final GA parameter is employed, none of the genetic variants studied affect it; instead, it is the baseline glycemic concentration, eGFR, and $Css^{max}$ with a good value of adjusted R Square of 0.727 that influence GA. In addition, changes in GA are affected by the baseline glycemic concentration, eGFR, and $Css^{max}$. Such findings on the effects of rs2289669 *SLC47A1* polymorphisms are different from those of other studies that use another parameter of glycemic response in metformin use. Research on the effects of rs2289669 on glycemic response to metformin using HbA1c reveals that the AA homozygous variant has the best glycemic response. This is probably caused by the reduced function of MATE1, which has an important role in the renal secretion of metformin, marked by a high AUC but low $Cl_R$ among ten patients with such variant as opposed to those with other variants [18]. This result is similar to that of the research on 142 patients in Slovakia in which 20% of those with AA homozygous variant have two-fold reduced HbA1c after using metformin for six months [41]. Another similarity is found among 116 Caucasian patients with T2DM where those with the A allele *SLC47A1* rs2289669 have 0.3% more reduction in HbA1c upon taking metformin [42]. Therefore, along with the baseline glycemic value and eGFR, the pharmacokinetics of maximum steady-state concentration (Cssmax) has a correlation with GA. The results of $Css^{max}$ examination in this study indicate that 64.6% patients have metformin concentrations in the therapeutic range (0.75–5 g/mL), and only 1/10 has $Css^{min}$ that is greater than or equal to 0.75 g/mL. This can possibly cause $Css^{max}$ to be the only parameter associated with GA. Therefore, these findings confirm the importance of adherence to metformin therapy to guarantee the achievement of metformin therapeutic concentrations.

Although the best efforts have been made through multicenter studies in some primary healthcare centers, there is a limitation in this study related to the number of patients involved. It becomes one of the factors in the incomprehensive analysis of the effects of polymorphisms on the pharmacokinetics of metformin steady-state concentrations and glycemic response. The difficulty in involving patients who are adherent to metformin therapy for a minimum of eight weeks is also a challenge for further studies.

## Conclusion

In general, this study has found that the three polymorphisms absolutely have no effects on the pharmacokinetics of metformin steady-state concentrations. Although a further analysis involving other variables indicate the influence of *SLC22A1* Met408del polymorphisms on the pharmacokinetics of metformin steady-state concentrations, the variables that are not studied here in fact play a more major role ($>95\%$). Alongside the baseline glycemic value, rs2289669 *SLC47A1* affects FBG while only eGFR and $Css^{max}$ influence GA, but the three SNPs studied do not. These findings lead to a recommendation of further studies involving more subjects for a safe approach of metformin dose, particularly in T2DM patients with the G allele *SLC22A1* Met408del to prevent metformin accumulation beyond the recommended therapeutic range.

## Acknowledgments

The authors express their gratitude to all the patients who participated in this study.

## Author Contributions

**Conceptualization:** Vitarani Dwi Ananda Ningrum, Ahmad Hamim Sadewa, Zullies Ikawati, Rika Yuliwulandari, M. Robikhul Ikhsan.

**Data curation:** Vitarani Dwi Ananda Ningrum, M. Robikhul Ikhsan, Rohmatul Fajriyah.

**Formal analysis:** Vitarani Dwi Ananda Ningrum, Ahmad Hamim Sadewa, Rohmatul Fajriyah.

**Funding acquisition:** Vitarani Dwi Ananda Ningrum.

**Investigation:** Vitarani Dwi Ananda Ningrum, M. Robikhul Ikhsan.

**Methodology:** Vitarani Dwi Ananda Ningrum, Ahmad Hamim Sadewa, Zullies Ikawati, Rika Yuliwulandari, M. Robikhul Ikhsan, Rohmatul Fajriyah.

**Project administration:** Vitarani Dwi Ananda Ningrum.

**Resources:** Vitarani Dwi Ananda Ningrum.

**Software:** Rohmatul Fajriyah.

**Supervision:** Vitarani Dwi Ananda Ningrum, Ahmad Hamim Sadewa, Zullies Ikawati, M. Robikhul Ikhsan.

**Validation:** Vitarani Dwi Ananda Ningrum, Ahmad Hamim Sadewa, Zullies Ikawati, Rika Yuliwulandari.

**Writing – original draft:** Vitarani Dwi Ananda Ningrum, Ahmad Hamim Sadewa, Rohmatul Fajriyah.

**Writing – review & editing:** Vitarani Dwi Ananda Ningrum, Ahmad Hamim Sadewa, Zullies Ikawati, Rika Yuliwulandari, M. Robikhul Ikhsan, Rohmatul Fajriyah.

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
