## [Decision Letter · Decision Letter 0]

25 May 2022

PONE-D-21-40655The Influence of Metformin Transporter Gene SLC22A1 and SLC47A1 Variants on Steady-state Pharmacokinetics and Glycemic ResponsePLOS ONE

Dear Dr. Ningrum,

I apologize for the delay in getting your manuscript promptly reviewed. As you might surmise, this was due to a lack of reviewers. Of the more than two dozen potential reviewers who were contacted, only one agreed to review. I am making an editorial decision to accept the manuscript with minor revisions based on the one reviewer's comments and my judgement of the manuscript.

While the study that you describe has a small sample size, it was conducted and analyzed well and is of relevance.

In your revision, please address all of the reviewer comments.

In addition, please see if the presentation can be improved; at some places, English grammar/usage can be improvised. As you know, PLoS ONE does not provide editorial service, and having a paper that is "professionally" prepared remains authors' responsibility. In table 2, comma instead of period is used as the decimal point indicator at some places. Please correct. % values are actually not noted in table 2. For all tables & figures, provide keys for the abbreviations/acronyms that are used.

We look forward to receiving your revised manuscript.

Kind regards,

Santosh K. Patnaik, MD, PhD

Academic Editor

PLOS ONE

Journal Requirements:

Reviewers' comments:

Reviewer's Responses to Questions

**Comments to the Author**

1. Is the manuscript technically sound, and do the data support the conclusions?

Reviewer #1: Partly

2. Has the statistical analysis been performed appropriately and rigorously? 

Reviewer #1: I Don't Know

3. Have the authors made all data underlying the findings in their manuscript fully available?

Reviewer #1: Yes

4. Is the manuscript presented in an intelligible fashion and written in standard English?

Reviewer #1: Yes

5. Review Comments to the Author

Reviewer #1: The authors aimed to investigate the influence of SLC22A1 rs628031 A>G (Met408Val) and Met420del genetic variants in OCT1 as well as SLC47A1 rs2289669 G>A genetic variant in MATE1 on the steady-state pharmacokinetics of metformin and the glycemic response.

As mentioned in discussion there is limitation of study that is the number of patients, so this study has not reached to any conclusion but leads to a path for further studies involving more subjects for a safe approach of metformin dose, particularly in T2DM patients with the G allele SLC22A1 Met408del to prevent metformin accumulation beyond the recommended therapeutic range.

Query1: At some places authors mentioned they have recruited T2DM patients taking metformin for eight weeks.

But some places e.g. in methodology in Measurement of the Glycemic Response: its mentioned The FBG and GA of T2DM patients given metformin monotherapy were measured before and after the continuous administration of metformin 500 mg twice daily for six weeks.

Please explain.

Minor concerns

Please give full form of MALA, eGFR.

Spelling error:

In introduction “Hipoglycemic”.

6. PLOS authors have the option to publish the peer review history of their article (what does this mean?). If published, this will include your full peer review and any attached files.

Reviewer #1: No

---

## [Author Response · Author response to Decision Letter 0]

28 Jun 2022

Vitarani D.A Ningrum

Department of Pharmacy

Universitas Islam Indonesia

Yogyakarta, Indonesia

June 29, 2022

Academic Editor

PLOS ONE

Thank you for your helpful comments. We have revised our paper accordingly and feel that your comments helped clarify and improve our paper. Please find below our responses:

1. Comments From Editor: 

Please provide additional details regarding participant consent. In the Methods section, please ensure that you have specified (1) whether consent was informed and (2) what type you obtained (for instance, written or verbal). If your study included minors, state whether you obtained consent from parents or guardians. If the need for consent was waived by the ethics committee, please include this information

These suggested edits have been completed (lines 108 – 117)

2. Comments From Editor: 

The presentation can be improved; at some places, English grammar/usage can be improvised

We have rechecked the grammatical accuracy of the manuscript

3. Comments From Editor: 

In table 2, comma instead of period is used as the decimal point indicator at some places.

These suggested edits have been completed (Table 2)

4. Comments From Editor: 

% values are actually not noted in table 2

 We have added % values in Table 2.

5. Comments From Editor: 

For all tables & figures, provide keys for the abbreviations/acronyms that are used 

 These suggested edits have been added (Table 1., Table 2., and Table 3.)

6. Comments From Reviewer: 

At some places authors mentioned they have recruited T2DM patients taking metformin for eight weeks.

But some places e.g. in methodology in Measurement of the Glycemic Response: its mentioned The FBG and GA of T2DM patients given metformin monotherapy were measured before and after the continuous administration of metformin 500 mg twice daily for six weeks.

Please explain

One of the inclusion criteria for this study is that the subject had been using metformin for a minimum of two weeks to ensure that the steady-state concentrations were achieved. A glycemic control examination was then performed at the beginning and six weeks after the routine use (the patient was confirmed to be compliant with the use of metformin during these six weeks). Therefore, in line with the reviewer's input, the author has revised the results and discussion section by adding such information (lines 106-107, 303-304)

7. Comments From Reviewer: 

Minor concerns

Please give full form of MALA, eGFR.

These suggested edits have been completed (eGFR line 40; MALA line 228) 

8. Comments From Reviewer:

Spelling error:

In introduction “Hipoglycemic”.

We have revised the word to hypoglycemia (line 61)

We appreciate your careful evaluation of our work and hope that this revision has addressed the comments and meets with your approval. We await your response.

Yours Sincerely,

Vitarani D.A Ningrum

---

## [Editor Report · Decision Letter 1]

30 Jun 2022

The Influence of Metformin Transporter Gene SLC22A1 and SLC47A1 Variants on Steady-state Pharmacokinetics and Glycemic Response

PONE-D-21-40655R1

Dear Dr. Ningrum,

We received your revised manuscript. The concerns raised in the last review have been adequately addressed, and I am pleased to inform you that your manuscript has been judged scientifically suitable for publication and will be formally accepted for publication once it meets all outstanding technical requirements.

Kind regards,

Santosh K. Patnaik, MD, PhD

Academic Editor

PLOS ONE
---

## [Editor Report · Acceptance letter]

13 Jul 2022

PONE-D-21-40655R1 

The Influence of Metformin Transporter Gene *SLC22A1* and *SLC47A1* Variants on Steady-state Pharmacokinetics and Glycemic Response 

Dear Dr. Ningrum:

I'm pleased to inform you that your manuscript has been deemed suitable for publication in PLOS ONE. Congratulations! Your manuscript is now with our production department. 

Kind regards, 

on behalf of

Dr. Santosh K. Patnaik 

Academic Editor

PLOS ONE